# Soft-Switching Full-Bridge Topology with AC Distribution Solution in Power Converters' Auxiliary Power Supplies

Salvatore Musumeci *[ID], Fausto Stella, Fabio Mandrile [ID], Eric Armando and Antonino Fratta

DENERG-PEIC Politecnico di Torino, Corso Duca degli Abruzzi 24, 10129 Torino, Italy;
fausto.stella@polito.it (F.S.); fabio.mandrile@polito.it (F.M.); eric.armando@polito.it (E.A.);
antonino.fratta@polito.it (A.F.)
* Correspondence: salvatore.musumeci@polito.it

**Abstract:** The auxiliary power supply in a power converter is a key topic in the optimization of the converter's low-voltage electronic circuit performance. In this article, a low-voltage DC-AC soft-switching full-bridge topology, with an innovative, driven technique to achieve a zero-voltage transition, is presented and discussed. The full-bridge converter drives a high-frequency transformer (called the main transformer) that on the secondary side, distributes an AC voltage and current to the several electronic circuit's supplies. Every power supply is composed of an HF transformer (called load transformer) that converts the AC secondary voltage of the main transformer to the voltage and current levels requested by the electronic circuit. In this paper, the operating conditions are first investigated by several simulation results. Furthermore, an actual DC-DC power converter is used as a workbench for an experimental investigation of the effectiveness of the proposed auxiliary DC-AC soft-switching topology, and the AC distribution approach, to realize the several points of load power supply requested. Finally, the advantages and drawbacks of this auxiliary power supply solution are critically discussed, providing guidelines for the power converter designer.

**Keywords:** auxiliary power supply; soft-switching DC-AC converter; zero-voltage transition; point of load supply; full-bridge MOSFET converter





## 1. Introduction

Nowadays, many efforts are directed at the development of innovative topologies of power converters with high efficiency and increasingly reduced in space, as well as enhanced switching device technologies, to improve reliability and performance [1–3]. A power converter is achieved by switching the topology composed of the power devices with associated gate drivers, passive components, such as capacitors, inductors, and high-frequency transformers. Furthermore, the control circuits are present in the power converter, formed of a mix of digital and analog circuits, with several input and output (I/O) ports to connect the converter system with the management environments. The need for increasingly compact converter equipment leads to design solutions with a closer arrangement of both electronic power circuits and signal boards. This proximity may be producing influences between the control signals and the power electrical quantities, which can affect the performance and the reliability of the power conversion system.

Two kinds of power supplies are required in the power conversion circuits. First of all, the main power source to supply the switching devices topology (in DC-DC converter, inverter system, controlled rectifier, etc.). Further, there are auxiliary power supplies for control, analog conditioning, sensing, and driver circuits. These auxiliary power supplies must be distributed to all the signal electronic control boards, without introducing further noise criticalities and failure conditions on the power converter system. The auxiliary power supply circuit is generally arranged with a power converter with multiple isolated outputs to power different electronic circuits [4]. In power supply circuits, a

crucial design criterion to avoid electrical noise is galvanic insulation and accurate layout design [5]. Auxiliary power supplies deserve a thorough investigation to optimize converter performance and reduce disturbance conditions to increase the reliability of the power converter. Furthermore, the different load requirements necessitate a careful design of the power supply at the point of use (local power supply—LPS) [6,7]. In some cases, dual voltage power supplies are required. In other operating conditions, sudden load variation, such as in the gate driver circuits, requires voltage stabilization, even under dynamic stresses [8]. Moreover, some loads need a strongly stabilized voltage to avoid errors in signal processing, such as in a microcontroller unit.

Another critical issue in an extended power converter system is related power distribution among the different subsystems that can be distributed on multiple electronic boards. The LPSs consist of isolated modular converter topologies. Modularity and sub-circuit isolation are required for reliability, functionality, and low susceptibility to interference. Furthermore, in electronic boards, there may be additional specific circuits with different power supplies and these are obtained through the point-of-load (PoL) solutions, which can be:

- switching non-isolated converters (Buck or Boost);
- isolated switching converters (Forward, Flyback or push–pull) [9];
- linear type, such as low voltage drop (LDO) regulators [10].

The number of power supply net rails necessary on a system's electronic board increases with the complexity of the mixed control and signal processing circuits that compose the powered loads. Several integrated modular switching converter solutions for the LPS arrangement are discussed in the literature and available in the market, with satisfactory efficiency and power density [11,12]. Moreover, the auxiliary power supply's rail distribution from the multiple output power supply to the electronic circuit's supplies requested is generally in DC.

Over the years, multiple solutions have been developed for insulated auxiliary power supply. The common solution presented in the literature, and used in many actual applications, is to have a main power supply circuit, such as a Flyback [13,14], or a Forward converter with multiple outputs [15]. Alternatively, soft-switching isolated converter solutions can be used to increase the efficiency and reduce EMI contents; however, they come with an additional number of components, an increased control complexity and a limited operational range [16,17]. Furthermore, despite the efficiency and the reduced EMI emissions, an additional figure of merit is represented by the scalability of the solution, e.g., in a conventional Flyback converter, adding a point of load would require the redesign of the Flyback transformer (coupled inductors).

In this article, a novel solution for distributed power supplies, for low-voltage circuits in a power converter, is presented and discussed. A soft-switching full-bridge silicon MOSFET converter is used as the main DC/AC isolated power supply, powering several LPS circuits through an AC bus distribution. An innovative driving strategy leads to a soft-switching zero-voltage commutation, allowing it to increase the overall converter efficiency, while reducing EMI emissions. Furthermore, the main full-bridge DC/AC topology features the full exploitation of the magnetic core of the insulation transformer, compared with a typical Flyback converter, since it acts on the entire hysteresis cycle through an AC signal. The isolated AC bus is distributed up to the point of load requested for the power supply of the electronic low-voltage circuits, thus, increasing system noise immunity (antenna effect is avoided). Each local power supply is supplied by a high-frequency (HF) load transformer that provides the input voltage level to drive the local voltage regulator, according to the requirements of the powered circuit [18]. The proposed approach allows for a simplified and isolated LPS circuit, with the use of a distribution of electrical quantities in AC.

Furthermore, an important advantage of the proposed solution is its scalability. It is possible to add an LPS without a redesign of the main HF transformer, as long as the maximum power ratings of the converter core are not overcome. The auxiliary power supply

solution is investigated and discussed by several simulation results and experimental tests. A DC-DC power converter is used as a workbench to demonstrate the effectiveness of the proposed low-voltage power supply arrangement.

This article is composed of the following sections:

- In the second section, the main arrangement and requirements of a distributed auxiliary power supply are investigated;
- In the third section, the soft-switching full-bridge converter topology and the operation conditions are analyzed. Furthermore, simulation runs are presented to describe and validate the operation of the main auxiliary converter;
- In the fourth section, an actual case of study is presented and the AC distribution solution for the auxiliary power supplies are considered.
- In the fifth section, the main converter benefits and the AC distribution are discussed, considering the advantages and the drawbacks compared with the traditional DC rails solutions.

## 2. Auxiliary Power Supply Arrangement and Load Request

In power converter topologies, the different low voltage electronic circuits need an appropriate power and voltage level, in dual or single-ended voltage solutions [19]. Furthermore, an insulation is necessary in several floating circuits, such as the gate drivers of the high-side switches. A commonly adopted architecture for the auxiliary power supply consists of a main DC/DC converter, interfaced with a DC source. An isolated DC/DC converter with multiple outputs can be adopted [20]. The converter power rate is in the range of 20–60 W. While the control or processing signal boards and gate drivers are powered by a specific LPS, every LPS converter output features dual or single voltage polarization, based on the loads' needs.

A typical distributed power architecture with DC sources and several LPS powering several loads at different voltages is depicted in Figure 1. The voltage and current outputs of the main DC/DC converter are distributed in DC [9].

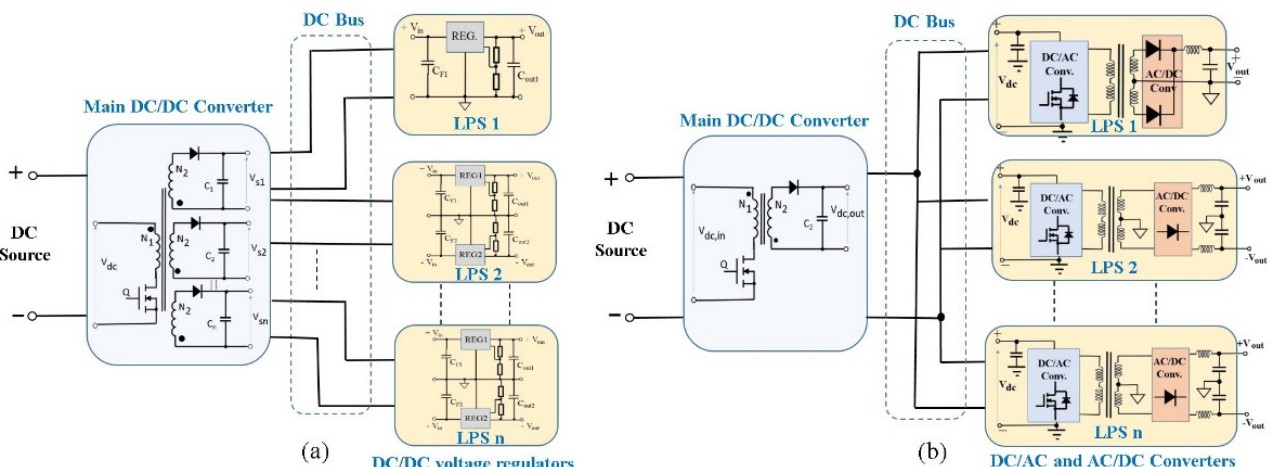

**Figure 1.** Typical distributed auxiliary power supply architecture with a main DC/DC converter with multiple output voltages and several LPS powering different kinds of low-voltage loads. (**a**) Multi-output isolated converter through a Flyback solution. (**b**) Single output DC/DC converter solution.

In Figure 1a, a multi-output isolated converter through a Flyback solution is considered. The multi-output solution (single magnetic core with several outputs) optimizes costs and saves components. However, it does not guarantee a fine regulation of all the load power supplies (especially if it has different voltage requests between the various channels) [21]. The multi-output solution requires a weightier design and a lower main DC/DC converter output voltage quality. Every LPS requires a linear LDO (or not isolated DC/DC switching converter). The single output DC/DC converter solution [22], depicted

in Figure 1b, distributes the single DC line to each load point. Each power supply stage is independently regulated and isolated. This auxiliary power supply solution offers a more modular approach (repeating non-isolated DC-DC blocks or isolated DC-AC and AC-DC converter blocks) and is aimed at obtaining enhanced quality of the output voltages of the various LPSs.

In terms of noise immunity, the multi-output solution features the disturbances limited to a single generation point (the main DC-DC converter). An input EMI filter is necessary [23]. On the other hand, the noise due to external disturbances propagates through the DC bus line. Therefore, additional galvanic isolation would be necessary to avoid noise propagation. Instead, the single voltage output solution (Figure 1b) generates disturbances in a distributed way, in the various power supply points of the power converter. In this second case, it is necessary to provide anti-interference filters for every LPS.

In the proposed solution, a soft-switching full-bridge converter is used to obtain an AC magnetizing current to drive an HF transformer. The AC output is connected to the local HF transformer to obtain several LPS circuits. The architecture of the auxiliary power supply system is described in Figure 2.

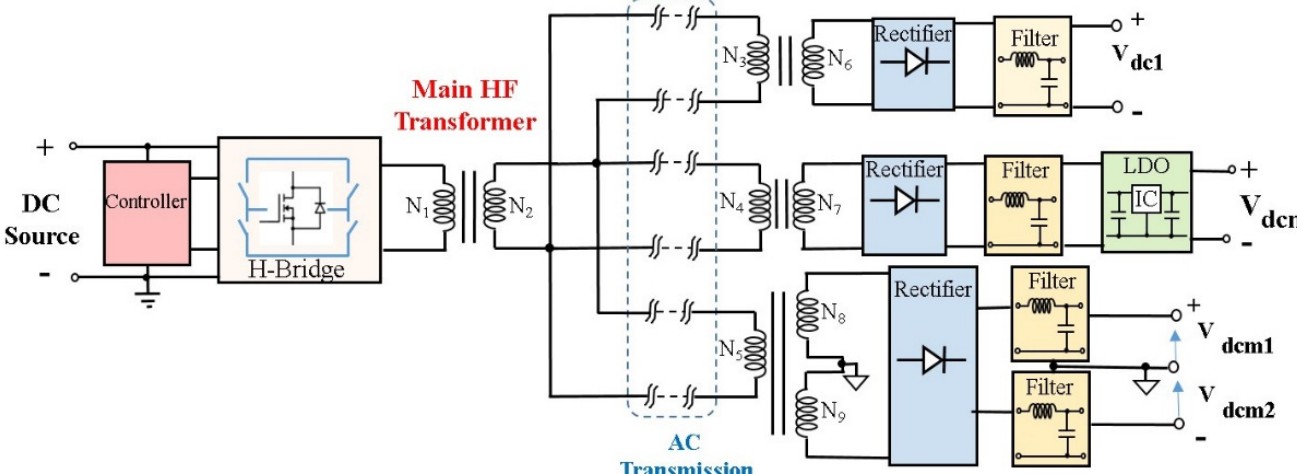

**Figure 2.** AC distributed auxiliary power supply's architecture with a main H-Bridge soft-switching converter and HF transformers to obtain the LPS requested low voltage ($V_{dcn}$) for the load's needs.

Each secondary circuit is composed of a fast rectifier diode bridge, followed by a low pass filter and a linear or switching regulator, if it is necessary to obtain a more stable and accurate output voltage. Further, in this LPS arrangement, the HF transformer can be with a single voltage on secondary or center-tapped secondary to obtain a dual-voltage output. The design of the low pass filter (yellow block in Figure 2) is crucial to supply the correct electrical quantities requested to the loads.

The LPS must satisfy the different needs of the loads. A simplified half-bridge topology, shown in Figure 3, is used to describe the main low-voltage electronic circuits that are necessary to supply.

The following main types of electronic circuits, powered by LPS, can be considered to investigate the necessary supply requirements:

- gate driver circuits;
- microcontroller circuits;
- sensing interface circuits;
- I/O circuits.

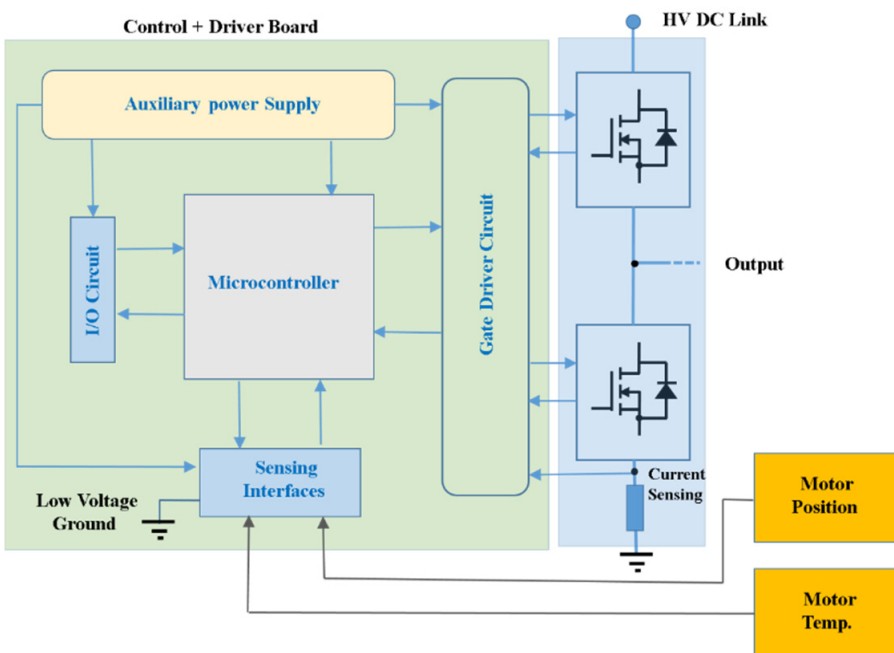

**Figure 3.** Half-bridge power converter with the main low-voltage electronic circuits that need auxiliary power supplies.

### 2.1. Gate Driver Circuit Power Supply Requirements

In power converters, silicon devices, such as MOSFETs and IGBTs, or wide-bandgap devices, such as SiC MOSFETs and GaN FETs, need to be driven with different voltages and currents levels. The power supply related to the high-side gate driver requires insulation from the ground. The voltage levels depend on the switching device. For example, in the case of SiC MOSFETs, the on voltage ($V_{GS,on}$) is in the range of 18–20 V, while the off voltage ($V_{GS,off}$) is generally negative, in the range of $-5$ V to $-2$ V [24,25]. The power supply must be capable of powering the gate drive of the device, considering the constraints of the gate charge [26]. Furthermore, the LPS power rate must include the whole driver circuit losses.

For insulated gate devices, the driving power losses are related to the gate charge $Q_G$ request, according to the equivalent device input model, shown in Figure 4 (depicted with a MOSFET, but the same consideration can be carried out for IGBT and GaN FET switches). The gate driver power losses are related to the gate charge $Q_G$, the gate voltage swing $\Delta V_{GS}$ (from positive to negative value) and switching frequency $f_{sw}$, as described in (1)

$$P_G = Q_G \cdot \Delta V_{GS} \cdot f_{sw} \tag{1}$$

where $Q_G$ is the gate charge of the driven insulated gate device and $f_{sw}$ is the switching frequency. In Figure 4, the different circuital paths, to drive both the turn-on and turn-off transients of the device, are highlighted. Furthermore, the positive and negative supply voltages needed in optimized converter switching leg operations are considered. From the simplified circuital model depicted in Figure 4, the power rate request of the driver circuit is achievable. An analysis procedure to achieve the driver power supply requirement is described in [27].

The LPS design must take into account these driver supply constraints for the correct sizing of both the core and the number of turns of the HF transformer. Furthermore, each type of gate driver supply side may be isolated, both on the high side and also on the low side, to avoid noise.

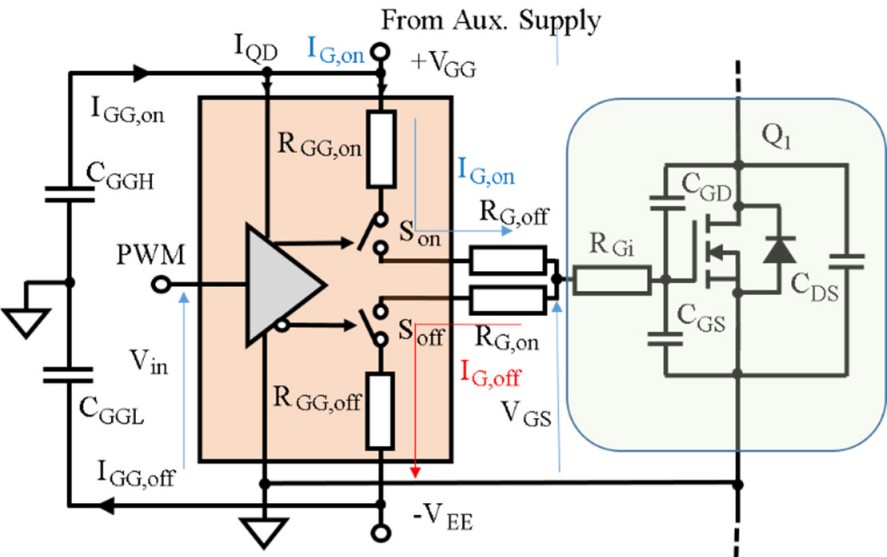

**Figure 4.** A simplified gate driver circuit and power MOSFET with inner resistance and capacitors model.

*2.2. Low-Voltage Control and Processing Circuit's Power Supply Requirements*

The use of microcontrollers, FPGAs (field-programmable gate arrays), and I/O circuits is widespread in power converter control boards. The device supplies are quite straightforward, but in industrial environments or in high-voltage converter applications, some trouble appears (voltage dips or glitch) due to the noises. The disturbance reduction needs galvanic isolation [28]. Furthermore, the low voltage requested by these electronic devices (3.3 V) requires step-down regulators, with satisfactory efficiency, limited output ripple, reduced size and cost. These design constraints require choosing a suitable power supply solution. A favorable way to step down voltage is by the use of a linear voltage regulator (LDO regulator) [12]. This is a practicable solution if the load current is relatively low and the efficiency is not a crucial design point. In this supply arrangement, the efficiency drops when the step-down voltage is very low, compared to that of the input voltage. In an LDO solution, the regulator dissipation can become a critical limitation at higher loads.

Higher efficiency may be achieved using switching step-down regulators.

If a switching converter is used as a voltage regulator, the inductor of the filter, downstream from the secondary of the HF transformer, is not strictly necessary. In sensing processing circuits (current and voltage, temperature, and so on), they mainly consist of analog circuits, which may also require dual voltages, obtained with two secondary turns in the HF transformer (as described in Figure 2). The LC filter can be followed or not by linear or switching voltage regulators, based on efficiency and performance needs.

Other global design considerations concern the layout of the printed circuit board (PCB). The layout, to be effective and less sensitive to disturbances, should be very close to the points to be powered (point of load) [6].

## 3. Soft-Switching Full-Bridge Converter Topology

The main DC/AC converter, described in Figure 2, is based on a soft-switching full-bridge MOSFETs converter that drives an HF transformer. The control operates in an open loop with 50% of the duty cycle. The schematic of the proposed power converter solution is reported in Figure 5a. In the schematic arrangement, an HF transformer with a center-tapped secondary is considered.

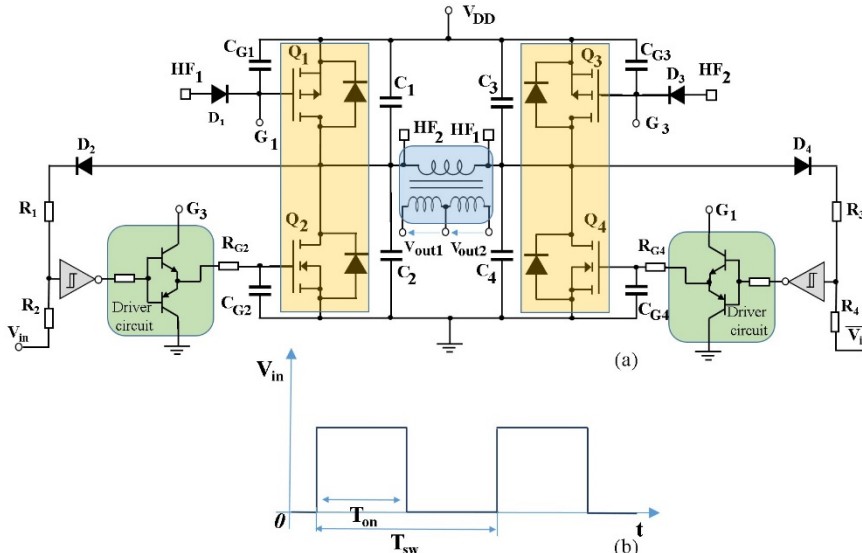

**Figure 5.** (**a**) Schematic of soft-switching full bridge with complementary power MOSFETs in the switching leg and innovative driving solution. (**b**) 50% of duty cycle PWM voltage control signal.

A square wave voltage is obtained by switching each half-bridge leg. The $dv/dt$ is regulated by the capacitive snubber, in parallel connection on every MOSFET.

The switching leg consists of two MOSFETs with the same power rating: a PNP MOSFET for the high side and an NPN MOSFET for the low side. This design choice allows for a strong reduction in the complexity of the driving circuit. The gate driving method and the capacitors in parallel connection across the drain and source of every device achieve the soft-switching operation. A simple oscillator circuit, operating in the range of 100 to 200 kHz, is used in a complementary way, to drive the two low-side devices of the switching legs, as shown in Figure 5b. The gate charging of the MOSFET devices takes a crucial role in the balancing conduction in the switching legs. The two resistor dividers ($R_1$–$R_2$ and $R_3$–$R_4$), and the diodes $D_2$ and $D_4$, are used to obtain the dead-time in the high-side and low-side devices of a switching leg. The resistive divider is designed to set the correct threshold ($V_{th}$) of the Schmitt trigger that drives the turn-on of the driven MOSFET device [29].

### 3.1. Power Converter Operation

The converter operation is divided into five stages. The general description of a switching leg voltage operative condition, with both the primary current and the magnetizing current, at a steady state, is depicted in Figure 6a. To describe the full-bridge converter operation (Figure 6b), a simple diode rectifier is considered in the secondary path, with a resistive load (Figure 6c).

In the first phase, it is assumed that $Q_2$ and $Q_3$ are turned ON and $Q_1$ and $Q_4$ are turned OFF; $V_{in}$ is zero while NOT $V_{in}$ is high. In this operating condition, the voltage at $HF_2$ is 0, while the voltage at the pin $HF_1$ is equal to $V_{DD}$. The current is flowing positive from $V_{DD}$ to the ground, through $Q_2$ and $Q_3$. Capacitors are in the following states: $C_1$ and $C_4$ are charged at voltage $V_{DD}$, while $C_2$ and $C_3$ are discharged at zero. The gate voltages $V_{GS,Q2}$ are high (about $V_{DD}$ minus BJT voltage losses) and $V_{GS,Q3}$ are low (zero voltage to ground, a negative voltage gate drain as $V_{GS,Q1}$ in state 3), while the $V_{GS,Q1}$ are high and $V_{GS,Q4}$ are low. State 1 is highlighted in Figure 7a. The switching conditions of MOSFETs in the full bridge are depicted in Figure 7b. The interaction between the primary inductor of the HF transformer and the capacitors' operative conditions is reported in Figure 7c.

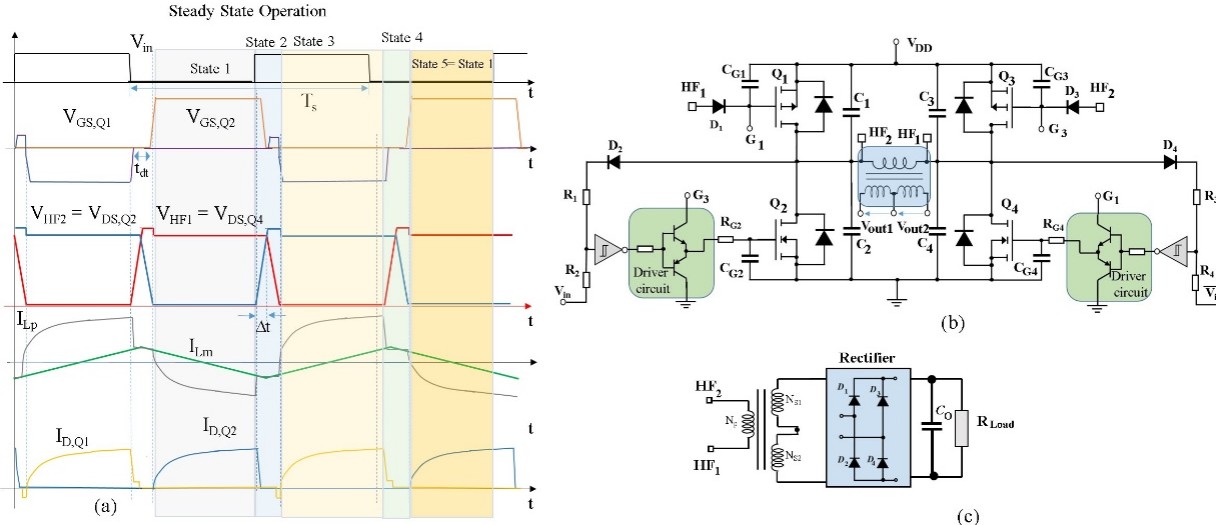

**Figure 6.** (**a**) Qualitative switching waveforms at steady state. (**b**) Schematic of full bridge based on complementary MOSFETs in switching leg. (**c**) Simplified transformer output circuit considered in the operation analysis.

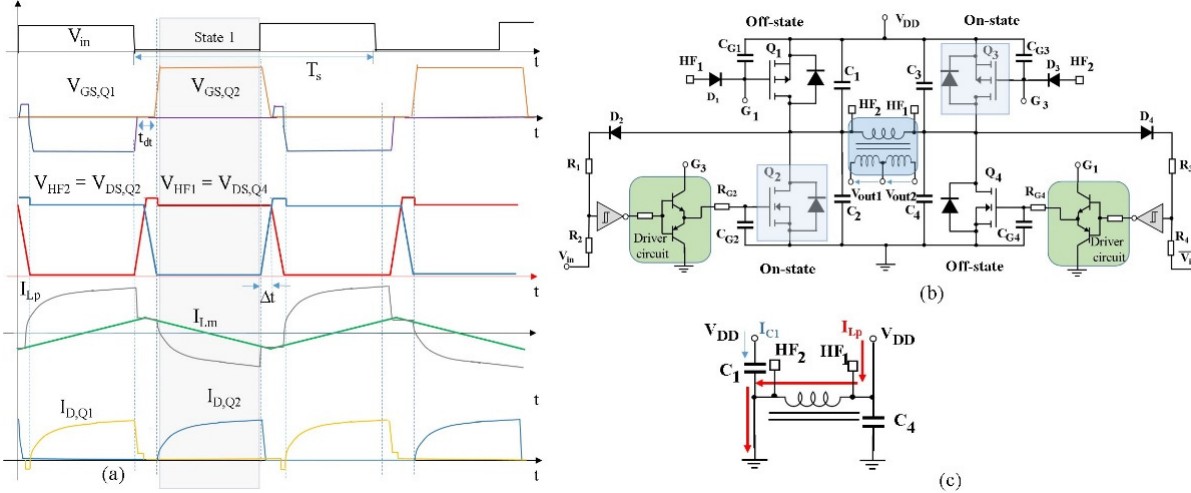

**Figure 7.** (**a**) Qualitative switching waveforms at starting of the description (State 1). (**b**) Switching conditions of the MOSFETs. (**c**) Primary side of the HF transformer and capacitors' interactions.

In the second phase, the first voltage transition occurs. $V_{in}$ rises high, while $V_{in}$ inverted, applied to the $Q_4$ driver circuit is 0. In the initial state, $V_{HF1}$ was high. The diode $D_4$ and the resistor divider, composed of $R_3$ and $R_4$, keep the voltage at the input of the Schmitt trigger high. Consequently, $Q_4$ remains off. By setting $V_{in}$ high, $Q_2$ is immediately turned off. The current in the magnetization inductance cannot go to 0 instantaneously, thus, it splits in the capacitors $C_2$ and $C_1$. The effect is to charge $C_2$ and discharge $C_1$ [29]

The operative conditions in the second phase of operation are reported in Figure 8.

Figure 8a describes the transient's state of the switching voltages and currents. Figure 8b shows the switching evolution of the MOSFETs and body diode of the high-side devices. The current and voltage behavior in the capacitors and in the primary-side inductor are depicted in Figure 8c.

In Figure 8a, looking at the magnetizing current of the transformer, it is at its peak value. With a good approximation, it can be considered constant during the switching transient phase.

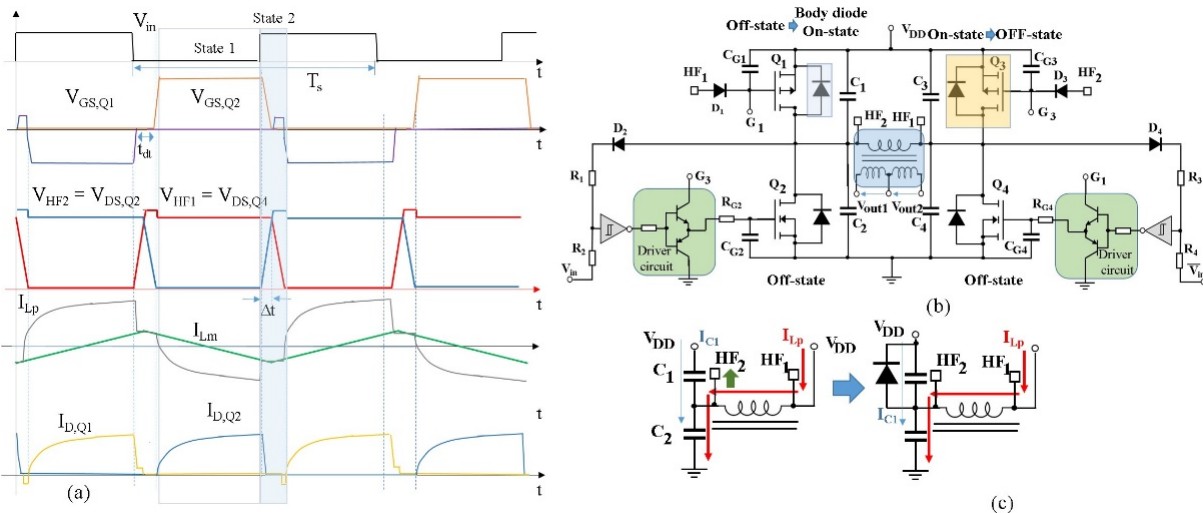

**Figure 8.** (**a**) Qualitative switching waveforms description at second phase of the operative condition (State 2). (**b**) Switching conditions of the MOSFETs. (**c**) Primary side of the HF transformer and capacitors' interactions.

The voltage peak reported in Figure 7a is due to the magnetizing current that turns ON the body diode of the upper device $Q_1$. The voltage $V_{DS,Q2}$, is clamped to the value of Equation (2):

$$V_{DS,Q2} = V_{DD} + V_F \tag{2}$$

where $V_F$ is the direct voltage drop of the body diode of $Q_2$.

The third phase is related to the second voltage transition, caused by $Q_3$ opening. When $Q_3$ is turned OFF, the voltage across the two capacitors $C_3$ and $C_4$ starts to change, with the same dynamic of the previous transition. The voltage $V_{HF1}$ decreases. When the voltage at the input of the Schmitt Trigger of the driver circuit of $Q_4$ decreases below the threshold value ($V_{th}$), $Q_4$ is turned ON. The driving circuit is made to connect, through the point $G_1$, the gate of the $Q_1$ and $Q_4$. In this way, the two devices are turned ON together. In Figure 9a the switching waveforms are depicted, highlighting state 3. The switching MOSFETs conditions are reported in Figure 9b. Finally, the primary side of the transformer-simplified model and capacitor interactions are depicted in Figure 9c.

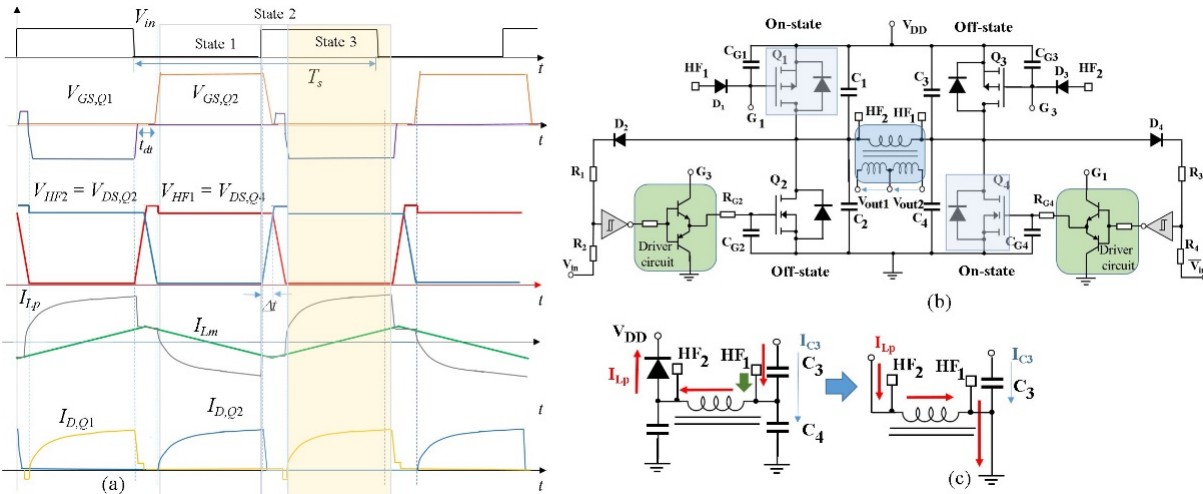

**Figure 9.** (**a**) Qualitative switching waveforms description at third phase of the operative condition (State 3). (**b**) Switching conditions of the MOSFETs. (**c**) Primary side of the HF transformer and capacitors' interactions.

The fourth phase is related to the third voltage transition, $V_{in}$ fall to 0 V. $\overline{V_{in}}$ goes high and $Q_4$ opens immediately. In the initial state, $V_{HF2}$ was high. The diode $D_2$ and the resistor divider, composed of $R_1$ and $R_2$, maintain the voltage at the input of the Schmitt trigger high, maintaining $Q_2$ as OFF. The voltage $V_{HF1}$ starts to rise with slope, depending on the $C_3$ and $C_4$. As we know, the current in the magnetization inductance cannot go to 0 instantaneously, thus, it splits in the capacitors $C_3$ and $C_4$, discharging $C_3$ on $C_4$. As in the second operating phase, the voltage on the HF1 pin exceeds the threshold voltage of the body diode of $Q_3$, resulting in a peak voltage in $V_{DS,Q4}$. The voltage high $V_{HF1}$ turns on the diode $D_1$ and connects the gate of $Q_1$ to the voltage $V_{HF1}$. $Q_1$ turns OFF at the end of the rise ramp of the voltage $V_{HF1}$. When $Q_1$ is turned OFF, the voltage across the two capacitors $C_1$ and $C_2$ starts to change, with the same dynamic of the previous transition described. The voltage $V_{HF2}$ decreases. When the voltage at the input of the Schmitt Trigger of the driver circuit of $Q_2$ decreases below the threshold value, $Q_2$ is turned ON. The driving circuit is arranged to connect, through the point $G_3$, the gate of the $Q_3$ and $Q_2$. In this way, the two devices are turned ON together and phase 1 returns. In Figure 10, the four operative conditions are described. In Figure 10a, the state 4 waveform behaviour is highlighted. In Figure 10b, the switching conditions of power MOSFETs and high-side body diode are focused on. Finally, in Figure 10c, the capacitors and primary-side inductor interaction and current behaviour are depicted.

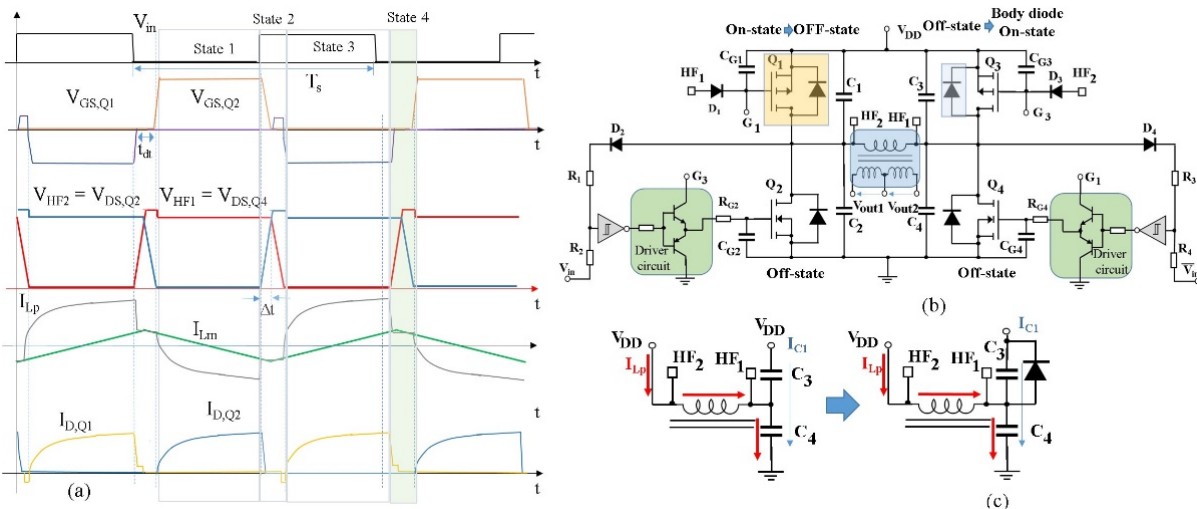

**Figure 10.** (**a**) Qualitative switching waveforms description at fourth phase of the operative condition (State 4). (**b**) Switching conditions of the MOSFETs and body diodes. (**c**) Primary side of the HF transformer and capacitors' interactions.

The fifth phase is similar to phase 1. The MOSFETs, $Q_2$ and $Q_3$, return in conduction and $C_1$ and $C_4$ start to charge, as shown in Figure 11. In Figure 11a, the state 5 curves are highlighted. In Figure 11b, the switching state of the MOSFETs are shown, and in Figure 11c, the current inversion on the primary inductor and the capacitors' charging involved in phase 5 are depicted.

### 3.2. HF Transformer Selection Issue

The correct HF transformer selection is crucial in the converter operation [30]. Considering a square wave voltage (i.e., waveform factor equal to 1), according to Faraday's law, the generated core magnetic flux ($\varphi_T$), in the core, features a triangular waveform, and as depicted in Figure 12, the maximum voltage can be written as:

$$V_{max} = N \cdot \frac{\varphi_{Tmax}}{\frac{T_{sw}}{4}} = 4 \cdot f_{sw} \cdot \phi_{Tmax} = 4 \cdot f_{sw} \cdot N \cdot A_e \cdot B_{Tmax} \tag{3}$$

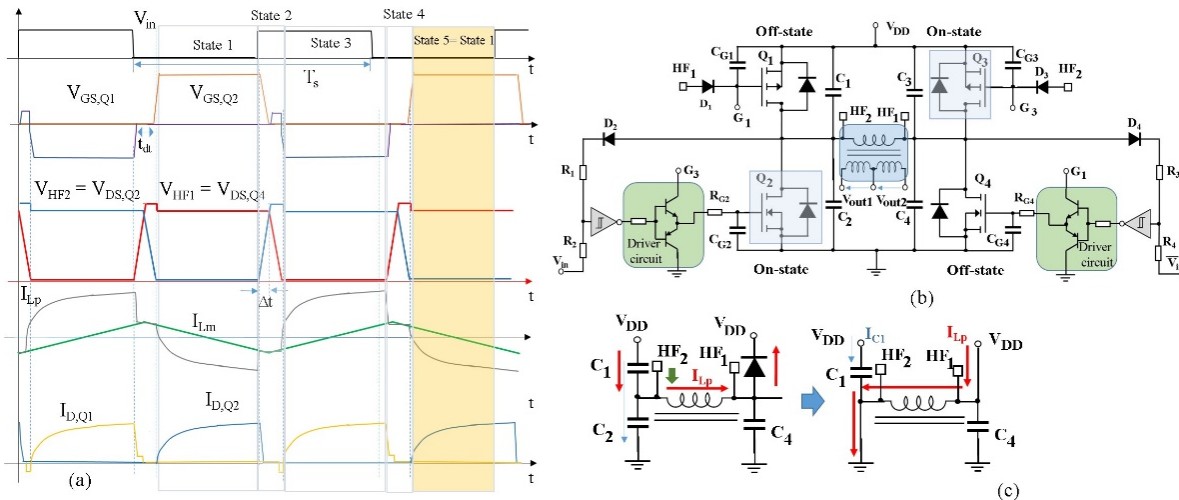

**Figure 11.** (**a**) Qualitative switching waveforms description at fifth phase of the operative condition (State 5). (**b**) Switching conditions of the MOSFETs. (**c**) Primary side of the HF transformer and capacitors' interactions.

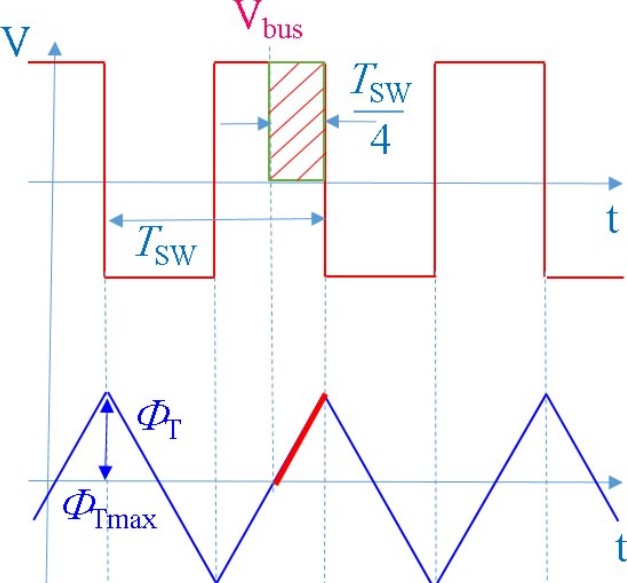

**Figure 12.** Qualitative voltage and flux density stress on the transformer. $\phi_T$ is the linkage flux and $\phi_{Tmax}$ is its maximum value.

$N$-turns are the transformer winding and $A_e$ and $B_{Tmax}$ represent the cross-sectional area of the core and the magnetic flux density, respectively, and $\phi_{Tmax}$ is the maximum linkage flux. This equation can be used to design the HF transformer.

Considering the data of commercial ferrite transformers, the V·μs data-sheet parameter (considering the saturation flux $\varphi_{sat} > \varphi_{Tmax}$) can be used to select a correct HF transformer by processing (3) and setting $V_{max} = V_{DD}$.

$$V_{DD} \cdot \frac{T_{sw}}{4} < N \cdot \varphi_{sat} \tag{4}$$

### 3.3. Notes on the Snubber Capacitor Selection

The voltage in the snubber capacitors increase in a time $\Delta t_i$ (where the index $i$ indicates the parallel generic capacitor $C_i$ with $i$ equal to 1, 2, 3, 4) with the following rate of rising.

$$\Delta t_i = \frac{2 \cdot C_i \cdot \Delta V_{Ci}}{I_{Lm}} \tag{5}$$

where $\Delta V_i$ is the voltage variation across the capacitor $i$ (equal to drain-source voltage of generic $Q_i$ device). The $I_{Lm}$ is the magnetization current; it is the difference between the current $I_{Lp}$ and $I'_{Ls}$

$$I_{Lm} = I_{Lp} - I'_{LS} \tag{6}$$

where $I_{Lp}$ is the current on the primary, while $I'_{LS}$ is the secondary current, reported to the primary side by the turns ratio.

The capacitors must satisfy the following design constraints:

- $C_i \gg C_{oss,i}$ (where $C_{oss,i}$ is the output inner capacitor of MOSFET, $i$ switches equal to $C_{DS}$ of Figure 4) must be imposed to control the $\Delta t_i$.
- To maintain the softness in switching turn-off transient, the following empirical relation can be used:

$$\frac{I_{Lm} \cdot t_{off}}{2C_i} \leq k, \text{ with } 1 \leq k \leq 2 \tag{7}$$

where $t_{off}$ is the turn-off MOSFET switching time and $I_{Lm}$ is the magnetizing current during the turn-off.

- An energetic criterion can be further considered:

$$\frac{1}{2} \cdot L_m \cdot I_{Lp}^2 \gg \frac{1}{2} \cdot 4C_i \cdot \Delta V_{Ci}^2 \tag{8}$$

where $\Delta V_{Ci}$ is the voltage variation across the capacitor $C_i$. $L_m$ is the HF transformer magnetizing inductor.

- Furthermore, the capacitors $C_i$ cannot be of too great value, in order to not increase the capacitor energy losses shown in (8).
- On the other hand, the capacitors $C_i$ cannot be too small so as to not go against the first criterion and to have a margin of adaptation of the circuit to compensate for the asymmetries of the magnetization current.

### 3.4. Power Converter Simulation Results

The full-bridge converter topology is analysed, considering a secondary path connected to a Schottky diode bridge rectifier with a filter capacitor and a resistive load.

The simulation results are carried out in LTspice, under the following conditions:

- $V_{DD}$ = 15 V,
- $C_1 = C_2 = C_3 = C_4$ = 10 nF,
- $C_{G1} = C_{G2} = C_{G3} = C_{G4}$ = 1 nF,
- $L_p = L_s$ = 20 μH

The simulation results are carried out on the simplified circuit, reported in Figure 13. The switching frequency is fixed to 100 kHz. In the secondary side of the HF transformer, a diode bridge rectifier with a filter capacitor is connected. A resistive load to vary the requested output power in the range of 5–30 W is considered.

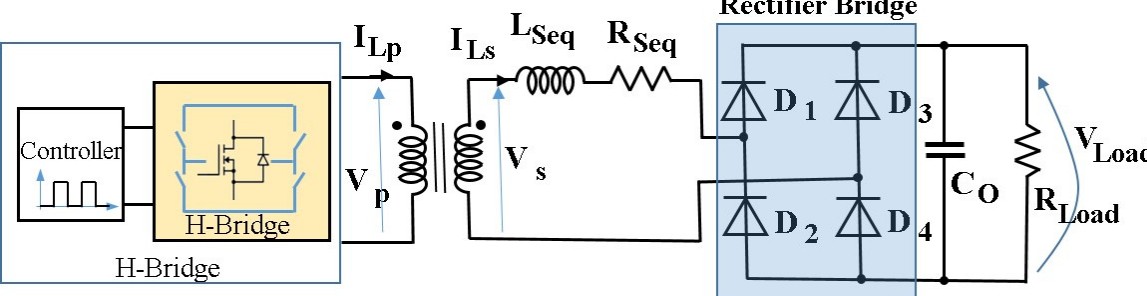

**Figure 13.** Simulation converter circuit and transformer secondary path arrangement for the simulation results.

The simulation results of the voltage waveforms, related to the gate voltage of the devices $Q_1$ and $Q_2$ node voltage of the $HF_2$ pin equal to the Drain Source voltage ($V_{DS2}$) of the low-side device $Q_2$, are reported in Figure 14. The secondary and primary voltage and current quantities at steady state are reported in Figure 15. The leakage inductance (in our case, an overall equivalent inductance of 100 nH in the secondary path is considered) affects the secondary current rise $I_{Ls}$, both in positive and negative transients.

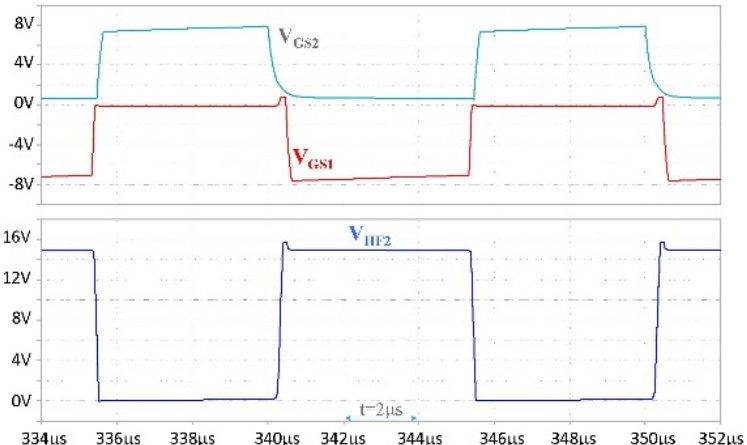

**Figure 14.** Simulation results in steady state of a switching leg. Up-side, the gate voltage of $Q_1$ ($V_{GS1}$) and $Q_2$ ($V_{GS2}$). Down-side the $V_{HF2}$ voltage corresponding to the $V_{DS2}$ across $Q_2$. Time t = 2 µs/div.

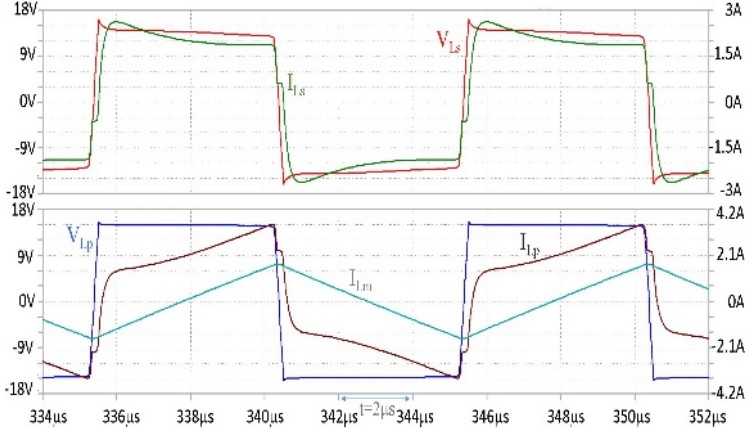

**Figure 15.** Simulation results in steady state. Up-side, the current and voltage across the inductor in the secondary side ($I_{Lp}$,$V_{Lp}$) with magnetizing current $I_{Lm}$. Down-side secondary side electrical quantities ($I_{Ls}$, $V_{Ls}$). Time t = 2 µs/div.

In Figure 16a, a zoomed-in view of the simulation results in steady state of the turn-off ZVS of device $Q_2$ is depicted. Furthermore, the snubber capacitor's currents are considered to show the magnetizing current is the sum of the absolute value of these two currents during the HF2 voltage rise (dv/dt). Moreover, as shown in Figure 16a, the voltage and current crossing is obtained at 1 V. The simulation waveforms are obtained considering a realistic value of equivalent parasitic resistance and inductance on the secondary side ($L_{seq}$ and $R_{seq}$), with a $V_{th}$ = 1 V and a resistive load ($R_{load}$) of 10 Ω. In Figure 16b, the same switching waveforms are carried out at heavy load. The capacitive currents show a noticeable current peak and the HF2 voltage curve, which features a double-voltage slope compared with the HF2 voltage of Figure 16a, due to the energy stored in the leakage inductance of the transformer.

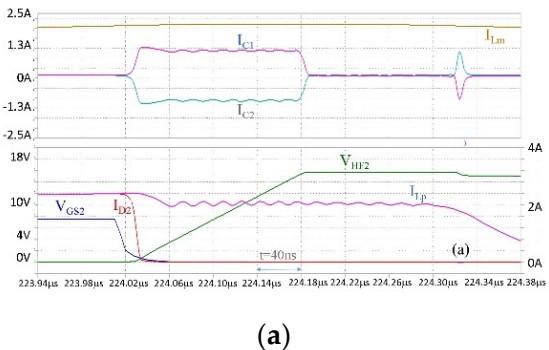

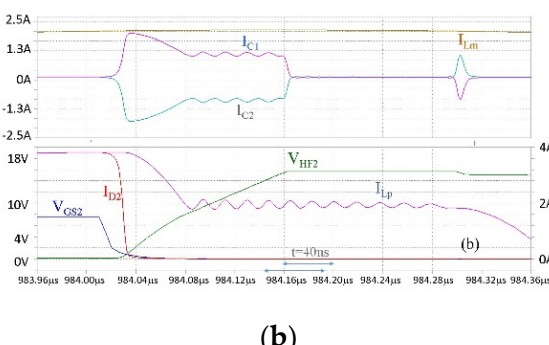

(**a**)                                                                                  (**b**)

**Figure 16.** Simulation results in steady state of the turn-off ZVS of device $Q_2$. In the up plot there are the magnetizing current $I_{Lm}$, the capacitors current $I_{C1}$ and $I_{C2}$. In the bottom plot there are the gate voltage of $Q_2$ ($V_{GS2}$), the $V_{HF2}$ voltage, the current on the primary side ($I_{Lp}$) the drain current of the MOSFET $Q_2$ ($I_{D2}$), time t = 40 ns/div. (**a**) Low load. (**b**) High load.

From inequality 7, it is possible to evaluate the capacitor value across the MOSFET.

In Figure 16a, the turn-off is evaluated in $t_{off}$ = 10 ns, the $I_{Lm} \approx 2$ A, considering 1 V as crossing voltage with the drain current $C_i$ = 10 nF.

## 4. AC Distribution for Auxiliary Power Supplies

In this section, the high-frequency soft-switching converter is used as a core of the auxiliary supply system, to power the several supplies dedicated to the driver circuits, the current sensing circuits, the conditioning and control circuits. The described auxiliary supply technique is investigated on the application in a main DC-DC power converter, evaluating the benefits and the limits.

### 4.1. A Case of Study

The soft-switching HF converter operation is experimentally validated in an actual 25 kW, [31] (800 V of DC bus) SiC MOSFETs-based DC-DC full-bridge converter, devoted to the first stage of an isolated charger circuit [32]. The simplified schematic of the power converter is reported in Figure 17a. The main HF open-loop soft-switching converter, with several LPS circuits, is highlighted in the schematic of Figure 17a. The picture of the power DC-DC converter and the auxiliary power supply circuits are depicted In Figure 17b. The HF soft-switching converter is based on an integrated half-bridge with an N and a P channel MOSFET (FDD8424 H). The main electrical parameters of the half-bridge MOSFETs are reported in Table 1. The capacitor across the drain-source of the MOSFETs, to obtain the soft-switching operation, is fixed to 10 nF and $V_{TH}$ is equal to 2 V. The main transformer selection procedure and characteristic parameters are reported in Appendix A. The DC bus is set to 15 V.

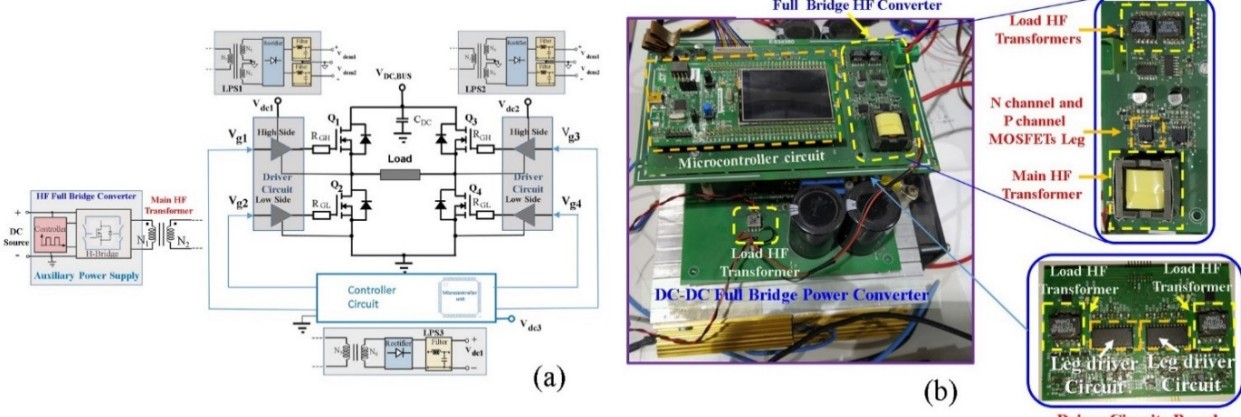

**Figure 17.** (**a**) Block schematic of DC-DC power converter and auxiliary power supply's arrangement. (**b**) Picture of experimental converter DC-DC power converter and auxiliary power supplies arrangement.

**Table 1.** Main power MOSFETs parameters.

| MOSFET Parameters | N-Channel | P-Channel |
|---|---|---|
| $R_{DSon}$ (@\|4.5\| V) | 23 m$\Omega$ (@ 7 A) | 58 m$\Omega$ (@ 5.6 A) |
| $V_{GS,TH}$ | 3 V | −3 V |
| $Q_G$ | 14 nC | 17 nC |
| $C_{iss}$ | 1000 pF | 1330 pF |
| $C_{rss}$ | 115 pF | 115 pF |
| $C_{oss}$ | 155 pF | 185 pF |

The switching waveforms of the main primary-side voltage at no-load on the HF1 and HF2 point of Figure 5a, with the primary voltage and the magnetizing current, are reported in Figure 18. In Figure 19a, the low-side MOSFET $Q_2$ gate-source voltage, with the drain-source voltage switching cycle in the no-load condition are shown. The zoomed-in view of zero-voltage transient operations is focused on in Figure 19b.

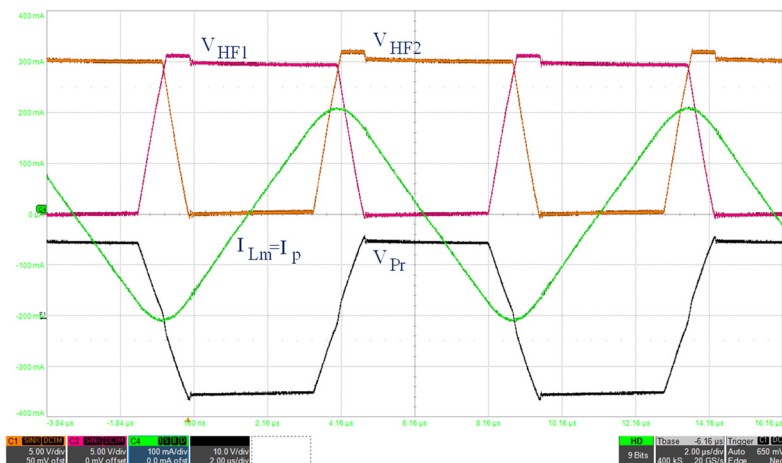

**Figure 18.** Switching waveforms at the primary side of the main HF transformer at no-load operation. $V_{Pr} = V_{HF1} = V_{HF2} = 5$ V/div, $I_{Lm} = 100$ mA/div, t = 2 μs/div.

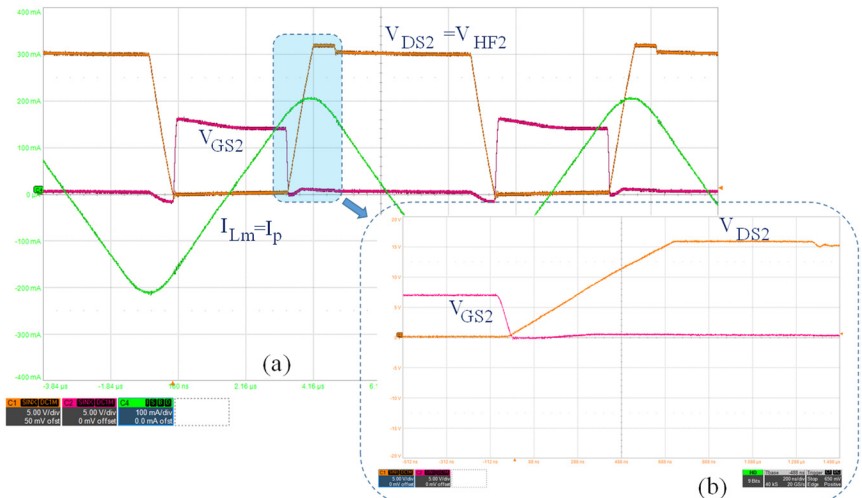

**Figure 19.** (**a**) Switching waveforms of gate-source voltage and drain-source of low-side MOSFET with the magnetizing current of the main HF transformer. (**b**) Focus on zero-voltage switching waveforms operation. $V_{DS2} = V_{GS2} = 5$ V/div, $I_{Lm} = 100$ mA/div, (**a**) t = 2 µs/div, (**b**) t = 200 ns/div.

The switching waveforms at load operative conditions are reported in Figure 20 ($I_{load} = 1$ A). While the gate-source voltage and drain-source voltage of the low-side ($Q_2$ MOSFET) device, at several load currents, are depicted in Figure 21a. The commutation voltage has a double slope, due to the energy stored in the leakage inductance of the transformer. During the first phase (higher slope) of the commutation, the leakage inductance keeps forcing current into the load and, therefore, the current at the primary side of the transformers is the sum of the load current and the magnetizing current. In the second phase (lower slope), the current in the leakage inductance goes to zero and, at the primary side of the transformer, only the magnetizing current is present, whose value is independent from the load. When the current in the load is sufficiently high, the current in the leakage inductance is maintained during the whole commutation; therefore, only the high slope is observed. Furthermore, the primary-side current and drain-source voltage at no-load and at 600 mA of the load current operative conditions are reported in Figure 21b.

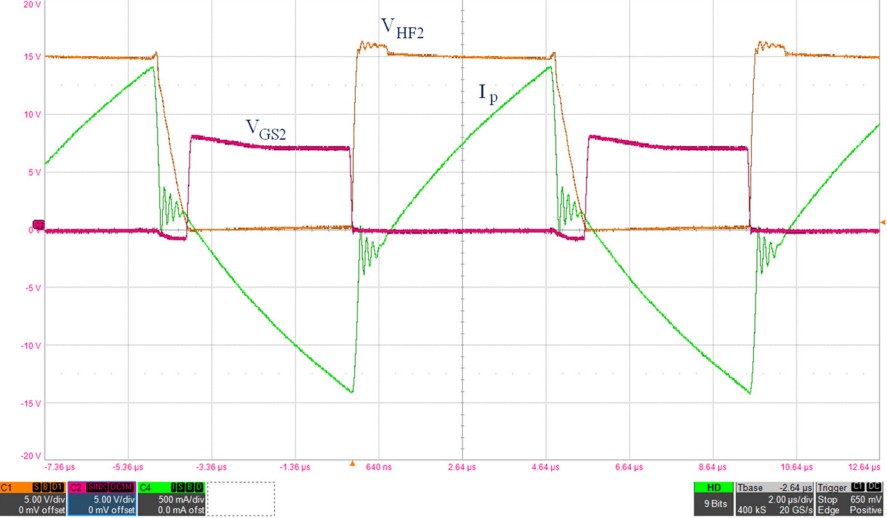

**Figure 20.** Switching Waveforms at high load ($I_{load} = 1$ A). $V_{GS2} = 5$ V/div, $V_{HF2} = V_{DS2} = 5$ V/div, $I_p = 1$ A/div, t = 2 µs/div.

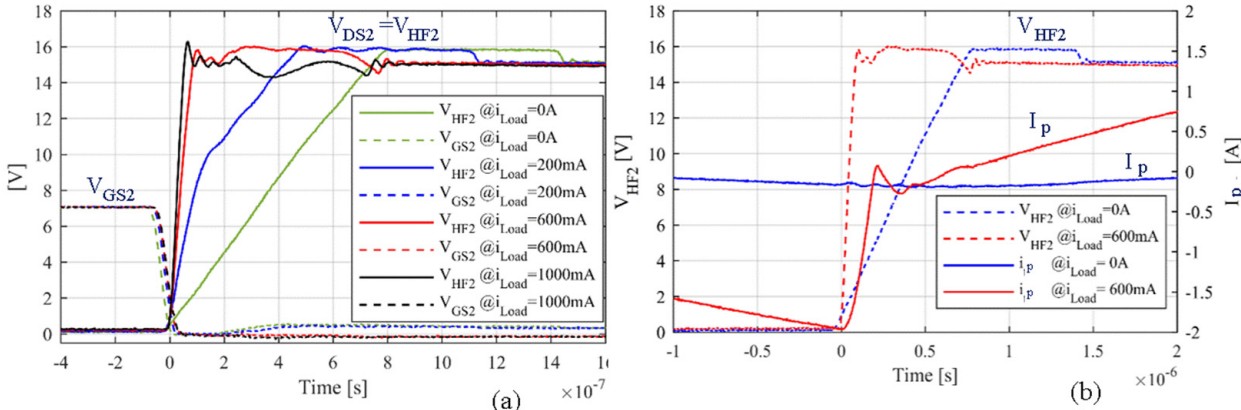

**Figure 21.** (**a**) Gate-source and drain-source switching transients of low-side device versus load current variation. (**b**) Primary-side current and drain-source voltage in two load current operative conditions.

The efficiency of the HF soft-switching converter is evaluated with the measurement circuit setup, depicted in Figure 22a, where the load variation is achieved by an electronic load. The efficiency considered is obtained by measuring the input power ($P_{in}$) with the power given at the secondary side ($P_{sec}$). This evaluation choice neglects the rectifier bridge power losses' contribution. The bridge diode rectifier is not considered in the efficiency because it is part of the secondary-side stage that depends on actual applications of the several electronic circuits served. The efficiency at different load currents is depicted in Figure 21b.

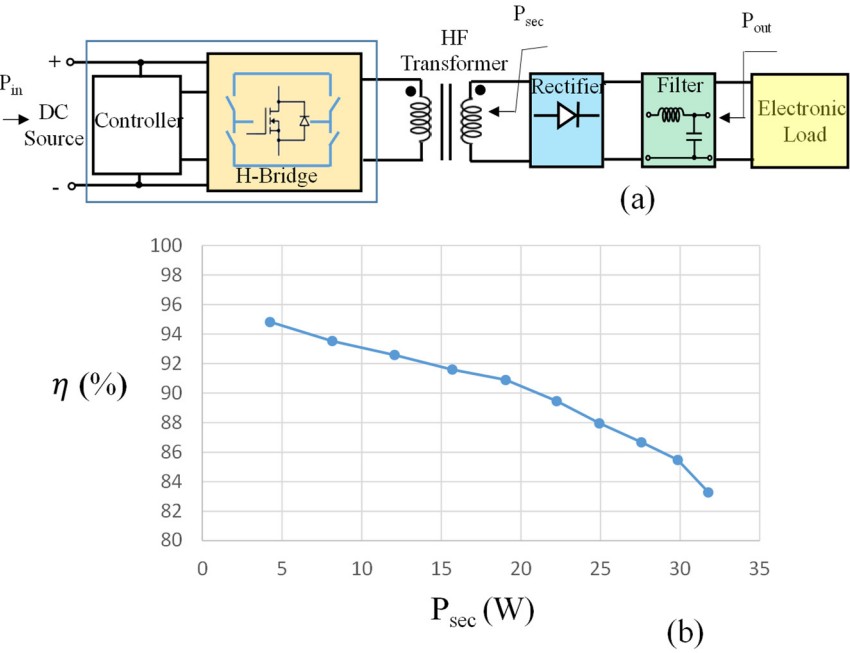

**Figure 22.** (**a**) Measurement circuit setup. (**b**) Efficiency versus power variation on the secondary path of the main HF transformer.

### 4.2. AC Distribution Experimental Evaluation

The load transformers allow for the supply of the requested voltage at several circuits, as depicted in Figure 17a. The secondary side of the main transformer is connected at different transformers in parallel connections. As a rule of thumb, the nominal power of the main transformer selected must be greater than or equal to the sum of the powers of the used transformers supplied. The load transformer "on site" manages the voltage and

current levels, based on the requests of the electronic circuit served. An actual arrangement and design of the load transformer application for the current sensing circuit is presented in Appendix A. The primary ($V_{pr,M}$) and secondary-side voltages ($V_{sec,M}$) of the main HF transformer with a secondary voltage of a load transformer ($V_{sec,L} = 15$ V) are reported in Figure 23a. The secondary-side waveforms of the load transformer for the positive voltage ($V_{sec,D}$) of the SiC MOSFET power device driver circuit is shown in Figure 23b. The DC-regulated voltage to power the driver circuit is also considered in Figure 23b.

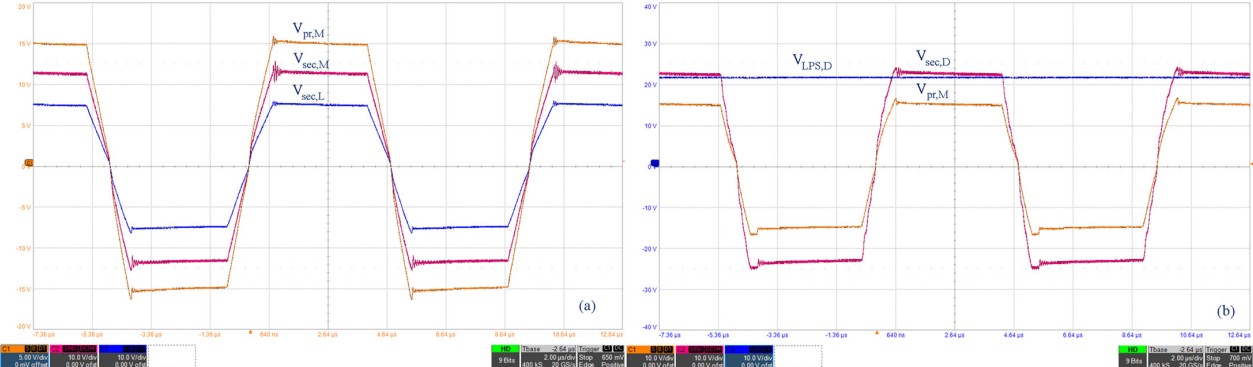

**Figure 23.** (**a**) Primary and secondary-side voltage of the Main HF transformer with a secondary voltage of a load transformer. $V_{pr,M} = 5$ V/div $V_{sec,M} = 10$ V/div, $V_{sec,L} = 10$ V/div (**b**) Primary-side voltage of the Main HF with the positive secondary voltage of load transformer devoted to the SiC MOSFET devices and the regulated DC voltage $V_{LPS,D}$. $V_{pr,M} = 10$ V/div $V_{sec,L} = 10$ V/div, $V_{LPS,D} = 10$ V/div, $t = 2$ μs.

### 4.3. Maximum Load Request

The zero-voltage commutation enables a strong reduction in the losses and the EMI emission. However, under extreme load conditions (e.g., low load resistance), soft-switching operations may not be guaranteed. During the switching operations, the snubber capacitors across the MOSFETs are charged and discharged using the energy stored in the magnetizing inductance of the transformers (main transformer + load transformer, in Figure 24a). In Figure 24b, the equivalent circuit of the converter is depicted, consisting of the main and load transformers, with the rectifier bridge at the secondary side of the load transformer, plus capacitor filter and resistive load. The energy stored in the transformers should be sufficient to charge/discharge the snubber capacitors. However, under heavy load conditions, during the voltage transition part, all the energy stored in the magnetizing inductance may go to the load, instead of to the snubber capacitors. This can cause a partial charge/discharge of the snubber capacitors, and if the threshold voltage necessary for switching is not reached, the converter stops. Supposing we have a purely resistive load, we can define it as 'limit resistance', the value of resistance beneath which the converter stops working. The limit resistance is affected by multiple variables, such as: the parameters of all the transformers (magnetizing inductance, leakage inductance, ohmic resistance), the voltage ripple on the load, the value of the snubber capacitances, the forward voltage of the diodes used in the bridge rectifier, the number of load transformers connected in parallel, and the threshold voltage selected for the soft commutation of the MOSFETs. As all these parameters must be considered, an analytical expression would be complex; therefore, the limit resistance has been computed by performing a SPICE simulation. Figure 24 shows part of the schematic of the case study circuit, simulated in SPICE, with the load transformer supplying a resistive load. The parameters of the transformers used are reported in Appendix A. Figure 25 shows the results from the SPICE simulation, where the parameters of the transformers are the same as the real case study.

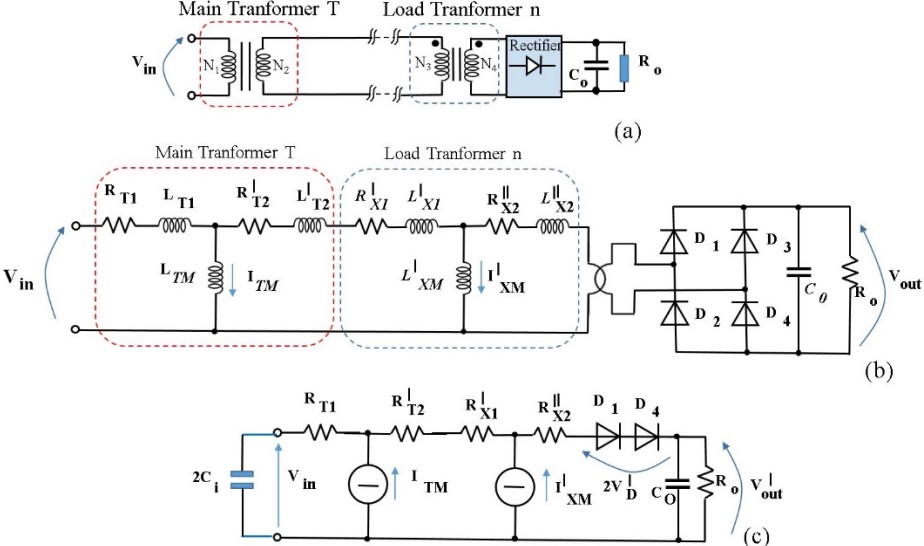

(a)

(b)

(c)

**Figure 24.** (**a**) Main transformer connected to the load transformer supplying a resistive load. (**b**) Equivalent transformer circuit for both main and load transformers reported in the primary side. The secondary side is connected with a full-bridge rectifier and capacitive filter which supply resistive load. (**c**) Simplified representation of the equivalent circuit of (**b**).

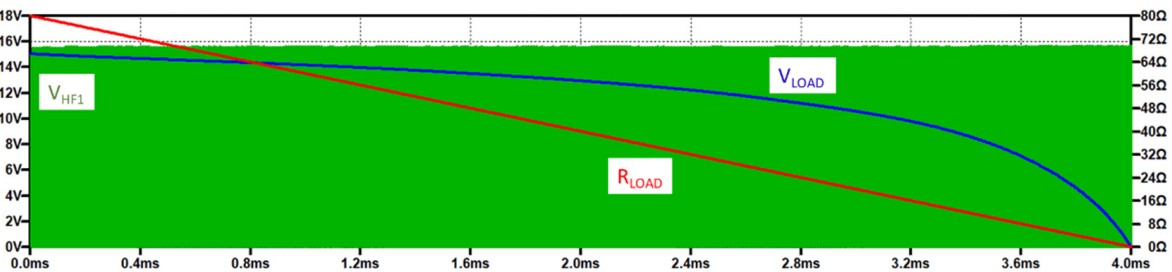

**Figure 25.** Load resistance lowered from 80 to 0 Ω. Case 1: high impedance transformer.

The green trace is the voltage at the node HF1, which alternates between 0 and 15 V, at a frequency of 100 kHz, while the blue trace is the voltage applied to the resistive load. The load resistance represented by the red trace is gradually lowered, from 80 Ω down to 0 Ω (i.e., the load current increases). In this case, the converter continues to work, also, when the load resistance goes to zero. This is mainly due to the load transformer that has high resistance (2.3 Ω at the primary side and 2.85 Ω at the secondary side) and high leakage inductance (6.5 µH at the primary and secondary side). In other words, the parasitics of the load transformer limit the maximum current in the load, and soft-switching operations are possible, also, when the load resistance drops to zero. A downside is that when the load current increases, the output voltage decreases significantly. However, this is usually not critical, as a voltage regulator (e.g., linear LDO) is commonly interposed between the diode bridge and the load. A second example is shown in Figure 26. In this case, the parasitics of the load transformer have been reduced, so as to have a resistance of 1 Ω on the primary and secondary side and leakage inductance of 1 µH on the primary and secondary sides. In this case, when the resistance of the load is below 7 Ω, the converter stops working, as it is not possible to perform the zero-voltage commutation. When the load resistance lowers under a certain limit, the magnetizing current that is normally used to charge/discharge the snubber capacitors goes in the load.

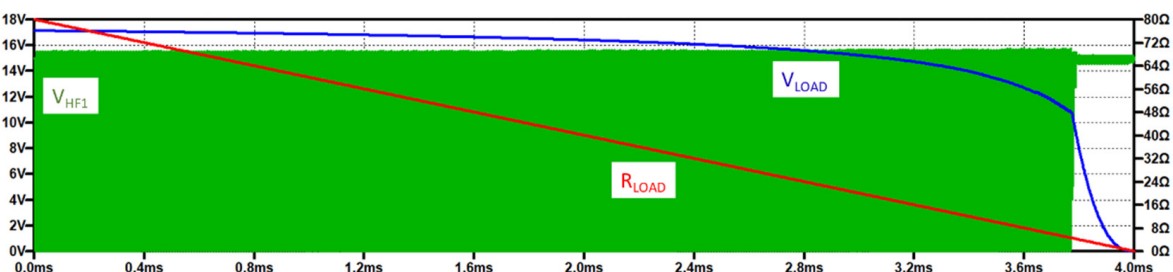

**Figure 26.** Load resistance lowered from 80 to 0 Ω. Case 2: low impedance transformer.

A second condition that can be critical is when the voltage ripple on the load is too high (e.g., insufficient load capacitance). In this case, during the commutation phase, the magnetizing current that, under normal operating conditions will charge/discharge the snubber capacitors, instead, flows in the load. That said, the snubber capacitors have to be charged up to the DC-link voltage, but if during the switching phase the voltage on the load drops considerably, then the magnetizing current finds a preferential path in the load.

As described, due to the complexity of the problem and the many variables to consider, it is not possible to provide a simple analytical expression for calculating the limiting resistance. However, as a rule of thumb for a good design, the parasitics of the main transformer should be at least one order of magnitude lower than the parasitics of all the other transformers and the voltage ripple on the load should be below 5%.

## 5. Discussion

The soft-switching MOSFET-based DC-AC converter presented can be applied as an auxiliary power supply in several kinds of power converters. The described DC-AC converter, through an AC electrical distribution, is able to power several analog conditioning and digital control circuits, as well as gate drivers, and so on. The auxiliary power converter circuit shows the following characteristics:

- Simple power-stage topology with an innovative soft-switching driving technique.
- The power rate of the full-bridge auxiliary converter can be designed to power several electronic circuits at the same time.
- Satisfactory efficiency of the auxiliary AC-DC converter, when the power required by the various loads is sufficient, thanks to the use of the zero-voltage soft-switching technique.
- Simplified open-loop control circuit, with 50% of the duty cycle on the full-bridge circuit, without the control capability of the HF transformer secondary voltages.
- The secondary side of the auxiliary DC-AC HF transformer is distributed into the different points of the power converter to supply the several requested low-voltage circuits.
- The electrical quantities distributed are in AC, from which the front-end device of every supplied circuit is another HF transformer (called load transformer).
- Every HF transformer achieves galvanic isolation on the electronic circuits powered, from which the noise rejection is improved.
- Easy scalability of the LPS arrangement. Additional LPSs can be easily added to the system, without the need to redesign the core H-bridge and main HF transformer (verified that the maximum power ratings of the converter are not overcome). The main HF transformer load limits are discussed in Section 4.3.

Several isolated DC-DC converters are available in the literature, among which, the Flyback converter is worth mentioning, due to its large use.

The main advantages of the proposed technology over the Flyback converter are:

- Lower EMI emission. Flyback converters are highly emissive due to the hard switching commutations that cause high dv/dt and di/dt.
- Lower switching losses. The proposed solution is soft switching (zero-voltage commutation), while the Flyback is hard switching in these kinds of applications.

- Lower breakdown voltage semiconductors. Flyback converter switches must be sized for a much higher voltage (input voltage + overvoltage at commutation + secondary voltage scaled by the Flyback transformer factor + safety margin), compared to the proposed solution (input voltage + safety margin) [33].
- Lower voltage ripple on the load. During the Flyback converter operations, the load is supplied only for part of the period, while in the proposed converter, the load is always supplied, except during the voltage transition. This results in a lower ripple voltage for the same load current and filter capacitance.
- Higher efficiency due to the soft-switching commutation.
- Better use of the magnetic component. The HF transformer magnetic cycle is on the fourth quadrant of the BH plane.
- The main advantages of the Flyback versus the presented topology are:
- Lower component count. A lower number of semiconductors and passive components.
- Widely used. Plenty of material is available in the literature and dedicated IC solutions are available on the market [34].
- Voltage regulation. The voltage on the secondary side can be regulated by acting on the duty cycle at the primary side.

As reported above, the proposed solution enables an important reduction in the EMI emission, compared to a classical hard-switching converter.

This is true not only for the emission generated during the commutation, where the soft switching avoids the generation of overvoltage and current oscillation, which are highly emissive. The second advantage is the fundamental voltage applied to the HF transformer that has a reduced number of high-frequency harmonic components. An example is shown in Figure 27, where the harmonic components of a voltage square wave ($V_{SQ}$), typical of a "standard" hard-switching converter, are compared with the harmonic components of the quasi-trapezoidal voltage wave, generated by the proposed converter ($V_{pr,M}$). Both FFT plots show the presence of the fundamental component at 100 kHz, plus a series of superior order harmonics. It can be seen that the proposed solution enables an important reduction in higher-order harmonics, especially in the range 500 kHz–10 MHz, thus, bringing important advantages in terms of EMI emissions. Furthermore, the transformer parasitics act as a low-pass filter, enabling a further reduction in the harmonic components of the AC voltage that is used to supply the load transformers.

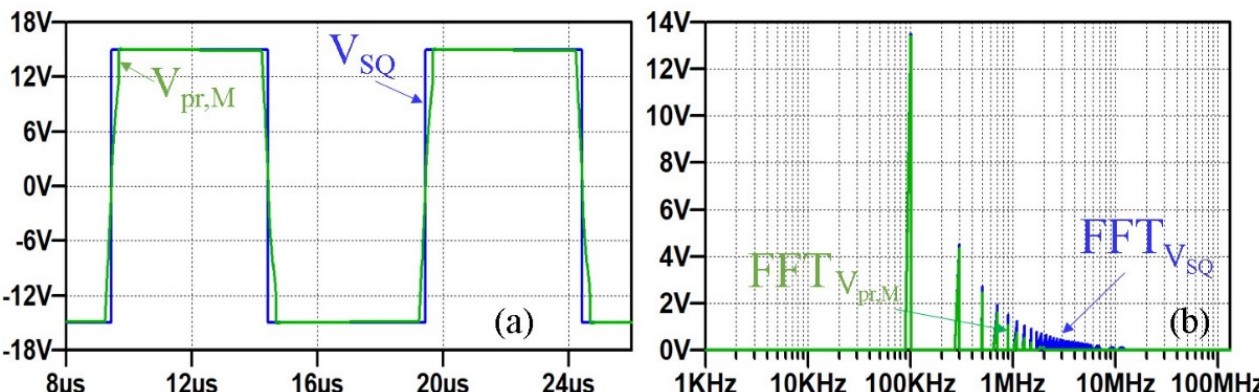

**Figure 27.** (**a**) square voltage wave ($V_{SQ}$) typical of a hard-switching converter compared with the primary voltage ($V_{pr,M}$) of the main HF transformer. (**b**) FFT of the voltage wave applied by the proposed converter to the primary side of the main transformer and the FFT of square wave voltage.

The proposed auxiliary power supply solution compared to a DC-distributed supply with local converters (one for each load, isolated or not), features a single main DC/AC bridge converter, followed by one transformer and one rectifier bridge for each load (with or without LDO regulator, based on the specific load request).

In contrast, a conventional solution with DC distribution (as described in Section 2) would require:

1. One DC/AC half or full-bridge solution, one transformer and one rectifier, for each isolated supply. Moreover, this would include additional filters for the EMI compatibility.
2. One DC/DC stage for each non-isolated supply

Therefore, the proposed solution requires fewer components and allows the optimization of the single main DC/AC bridge (soft-switching converter), therefore, maximizing the overall efficiency. This advantage becomes more significant in the case of multiple isolated supplies (such as gate drivers for multilevel structures [20]), as the number of DC/AC units can be reduced dramatically.

## 6. Conclusions

In this paper an innovative, auxiliary power supply solution, for power converter low-voltage electronic circuits, was presented and described. The power supply is based on a DC/AC soft-switching converter that drives a high-frequency transformer. The auxiliary power converter is based on an N-channel and P-channel MOSFET device, driven by an optimized driving technique, to obtain soft-switching transients. Simulation results were carried out to investigate the operation conditions and the limits of application. An experimental validation was presented in an actual case study, considering an auxiliary power supply for a 25 kW high-voltage full-bridge phase-shift DC-DC converter. The experimental results were carried out at several operative conditions, to demonstrate on the application, the effectiveness of the proposed solution. Furthermore, the satisfactory efficiency of the DC-AC stage with the HF transformer is described. The proposed soft-switching DC-AC converter, through the secondary path of the HF transformer, delivers by an AC bus, the electrical quantities in every point needing an auxiliary supply. At the point of load, an HF transformer (called load transformer) is used to adapt the main transformer secondary AC voltage level to the load request. A rectifier bridge with a suitable filter and a LDO (if necessary) is connected to the secondary path of the load transformer to power the requests of a DC load. This solution features an isolated power supply, very close to the load, improving noise rejection and, consequently, the converter robustness. The main DC/AC soft-switching converter shows an enhanced FFT, with a reduction in higher-order harmonics, compared with a hard-switching DC/DC isolated converter. Furthermore, quite a reduced number in overall components are achieved compared with a conventional DC/DC auxiliary converter, with DC distribution for local LPS. Finally, the proposed AC distribution system allows manageable scalability in adding LPS.

**Author Contributions:** Conceptualization, S.M., F.S. and A.F.; methodology, S.M., F.S. and F.M.; software, F.S. and F.M.; validation, E.A., F.S. and A F.; formal analysis, S.M., F.S. and F.M.; investigation, S.M. and A.F.; resources, E.A. and F.M.; data curation, S.M., F.S. and F.M.; writing—original draft preparation, S.M. and F.S.; writing—review and editing, F.M., E.A. and F.S.; visualization, E.A., F.S. and F.M.; supervision, E.A. All authors have read and agreed to the published version of the manuscript.

**Funding:** This research received no external funding.

**Conflicts of Interest:** The authors declare no conflict of interest.

## Appendix A

*Load Transformer Selection Example*

A power supply for the current sensing circuit is considered as an example of load transformer selection. The main HF transformer secondary path (see Figure A1a) is connected with two load transformers, to obtain the positive and negative voltage level. The two rectified voltages are regulated by an LDO circuit to achieve the requested supply's voltages (Figure A1b).

**Table A1.** Transformers Parameters.

| Part Number | Turn Ratio | Lm [μH] | Lσ1 [μH] | Rσ1 [Ω] | Lσ2 [μH] | Rσ2 [Ω] | Volt-Time Product [V∗μs] |
|---|---|---|---|---|---|---|---|
| Main Transformer | 1:1.55 | 158 | 0.5 | 0.014 | 0.8 | 0.022 | 443 |
| Load Transformer DA2099-AL_ | 1:1 | 3790 | 6.5 | 2.3 | 6.5 | 2.85 | 221 |

The power ratings of the main and load transformers, presented in the case study analysis, are reported in Table A1. The main transformer has considerably lower parasitics than the load transformer. In the case study, multiple load transformers (not presented here) have been connected in parallel. The main transformer must have an adequate power rating to supply all the load transformers. The main transformer of the case study has been custom designed and then experimentally characterized.

The first thing to verify is that the voltage-time product (flux) does not saturate the core of the transformer. The following relationship must be verified for each transformer (See Figure 12):

$$\phi_{Tmax} = V_{supply} \cdot \frac{T_{SW}}{4} < \phi_{sat} \tag{A1}$$

where $\phi_{sat}$ is

$$\phi_{sat} = N \cdot \varphi_{sat} \tag{A2}$$

In the commercial HF transformer, this value is achieved in the datasheet.

For the main transformer:

$$\phi_{Tmax} = 15 \cdot \frac{10 \text{ μs}}{4} = 37.5 \text{ V} \cdot \text{μs} < 443 \text{ V} * \text{μs} \tag{A3}$$

For the load transformer (DA2099-AL_), considering the output voltage of the main transformer (turn ratio 1:1.55):

$$\phi_{max} = 23.25 \cdot \frac{10 \text{ μs}}{4} = 58.125 \text{ V} \cdot \text{μs} < 221 \text{ V} * \text{μs} \tag{A4}$$

In case of saturation of the core, the switching frequency of the converter can be increased or a new transformer with a higher saturation flux can be selected.

Secondly, all the losses in the copper and the core of the transformers must be computed in the worst-case scenario (maximum load). The maximum temperature of the transformer must not overcome the maximum operating temperature recommended by the manufacturer; however, to maintain a high level of efficiency, the authors suggest limiting the temperature rise of the transformer to a few tens of degrees.

In the case study, the selected transformers have a considerably high saturation flux (V∗μs product) compared to the one that is strictly needed. This is due to the limited choice of transformers (for the off-the-shelf load transformer) and cores (for the custom-designed main transformer) available on the market. Furthermore, operating the transformers at a lower flux enables one to reduce the core losses and to increase the efficiency of the system.

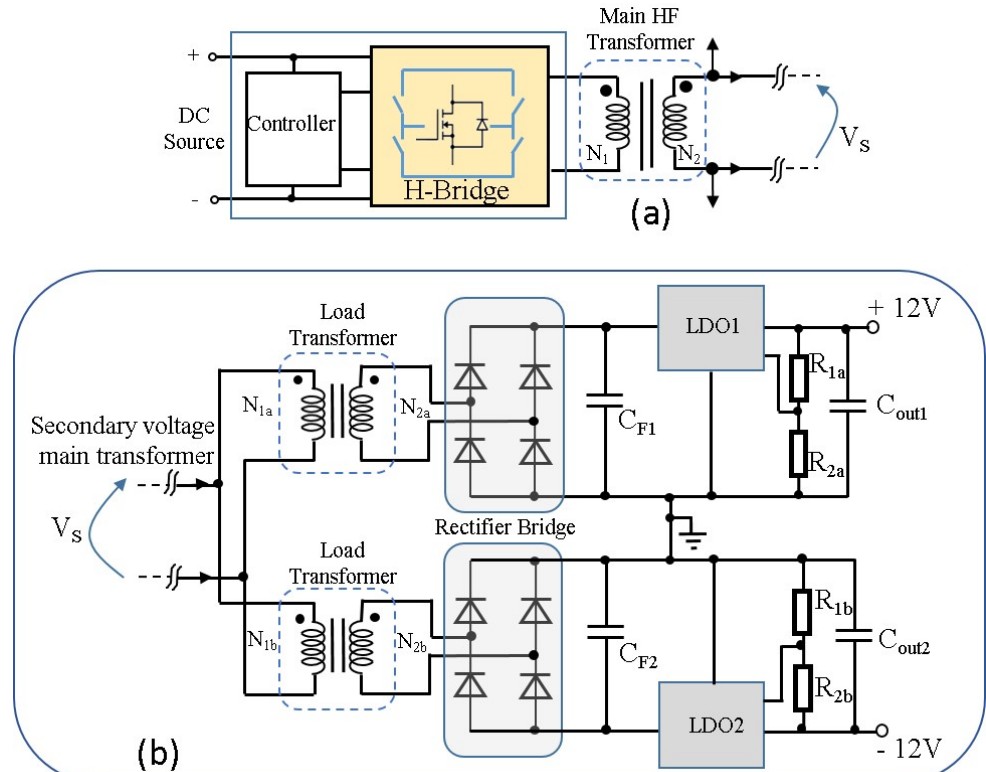

**Figure A1.** (**a**) The soft-switching auxiliary converter with the main HF transformer. (**b**) Current sensing power supply with the two load transformers for the positive and negative voltages source request.

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
