# Peer review of "Soft-Switching Full-Bridge Topology with AC Distribution Solution in Power Converters’ Auxiliary Power Supplies"

_electronics, doi:10.3390/electronics11060884_

Round 1

Reviewer 1 Report

The article presented by the authors is interesting, but still needs some improvement.

Shouldn't the waveform factor be included in formula 3? The waveform factor for a square wave signal with 50% duty cycle is 1, but maybe it's worth mentioning in the article?

Many drawings are illegible. In order to increase readability, it is worth enlarging them, e.g. Figs. 6b, 7b, 8b, 9b, 10b, 11b.

A few figures with waveforms require enlarging the x and y axis description fonts - they are illegible, e.g. fig. 14, 15, 16, 18, 19, 20, 23. 

There is no consistency in markings in the text and figures. Some variables are written in italic and others are written straight. These variables need to be standardized throughout the article, in the text and in the drawings.

Why is the formula 8 written in Bold (the variables are not vectors).

On line 94 there is "ant" and it should be "and".

There are two dots in line 212. 

On line 611 there is "he" and it should be "The".  

Please quote 1-2 articles from the mdpi journal.

Author Response

Review1

The article presented by the authors is interesting but still needs some improvement.

The authors thank the reviewer for the interest shown in the topic covered in the article and for the valuable suggestions that made the paper more effective.

Shouldn't the waveform factor be included in formula 3? The waveform factor for a square wave signal with a 50% duty cycle is 1, but maybe it's worth mentioning in the article?

Thank you for the comment. We clarified this in the revised paper.

Many drawings are illegible. In order to increase readability, it is worth enlarging them, e.g. Figs. 6b, 7b, 8b, 9b, 10b, 11b.

A few figures with waveforms require enlarging the x and y axis description fonts - they are illegible, e.g. fig. 14, 15, 16, 18, 19, 20, 23. 

Done. We enlarged and adjusted. All the figures.

There is no consistency in markings in the text and figures. Some variables are written in italic and others are written straight. These variables need to be standardized throughout the article, in the text and in the drawings.

We fixed it and now they are standardized.

Why is the formula 8 written in Bold (the variables are not vectors).

On line 94 there is "ant" and it should be "and".

There are two dots in line 212. 

On line 611 there is "he" and it should be "The".  

We fixed the formatting issues and the typos.

Please quote 1-2 articles from the MDPI journal.

We expanded the references by adding the following MDPI Articles:

1) Medina-Garcia, A.; Schlenk, M.; Morales, D.P.; Rodriguez, N. Resonant Hybrid Flyback, a New Topology for High-Density Power Adaptors. Electronics 2018, 7, 363. https://doi.org/10.3390/electronics7120363.

2) Darbas, C.; Olivier, J.-C.; Ginot, N.; Poitiers, F.; Batard, C. Cascaded Smart Gate Drivers for Modular Multilevel Converters Control: A Decentralized Voltage Balancing Algorithm. Energies 2021, 14, 3589. https://doi.org/10.3390/en14123589Rice J.;

3) Tahan, M.; Bamgboje, D.O.; Hu, T. Compensated Single Input Multiple Output Flyback converter. Energies 2021, 14, 3009. https://doi.org/10.3390/en14113009.

4) Leng, C.-M.; Chiu, H.-J. Three-Output Flyback Converter with Synchronous Rectification for Improving Cross-Regulation and Efficiency. Electronics 2021, 10, 430. https://doi.org/10.3390/electronics10040430.

Reviewer 2 Report

The technical novelty of this work should be detailed. Authors are required to better highlight the contributions of the work. The literature review has to be improved significantly and authors have to review more recent works and highlight the contribution of the work with respect to them. It is recommended that authors use other state-of-the-art methods for the sake of comparison. The paper writing requires a careful review for correcting grammatical mistakes and typos. More performance measures have to be used to evaluate the proposed work and other methods. The validation is a major issue of this work. How authors can avoid bias prediction? There are many figures in this paper, some figures are vague, the manuscript's figures need a brush-up.  

Author Response

Review2

The authors thank the reviewer for the valuable suggestions and corrections that helped improve the overall quality of the paper.

The technical novelty of this work should be detailed. Authors are required to better highlight the contributions of the work. The literature review has to be improved significantly and authors have to review more recent works and highlight the contribution of the work with respect to them. It is recommended that authors use other state-of-the-art methods for the sake of comparison. 

We added some considerations in the introduction and in the second section (Auxiliary Power Supply Arrangement and Load Request). We pointed out the benefits of the proposed converter compared to the state-of-the-art solutions that are currently used for addressing the same problem.

Figure 1 is changed to better describe the state of art and issues that lead to the proposed soft-switching converter and AC voltage and current distribution for the local power supply (LPS). Furthermore, several references are added to improve the literature review on this topic.

More performance measures have to be used to evaluate the proposed work and other methods. The validation is a major issue of this work. How authors can avoid bias prediction? 

We expanded the Introduction and the discussion of the results to highlight the pros of the proposed solution. In particular, we focused on the scalability of the proposed AC distribution solution. Furthermore, the performances in terms of harmonic behavior have been exploited in the final discussion The FFT of the trapezoidal voltage applied by the proposed converter to the primary side of the main transformer has been compared with the FFT of a square wave voltage typical of a classical hard switching converter.

In particular, the following sentences have been added to better understand the benefit in terms of component reduction

“The proposed auxiliary power supply solution compared to a DC distributed supply with local converters (one for each load, isolated or not), features a single main DC/AC bridge converter followed by one transformer and one rectifier bridge for each load (With or without LDO regulator, based on the load request).

While a conventional solution with DC distribution (as reported in section 2) would require:

  • One DC/AC half or full bridge solution, one transformer, and one rectifier for each isolated supply. Moreover, this would include additional filters for the EMI compatibility.
  • One DC/DC stage for each non-isolated supply

Therefore, the proposed solution requires much fewer components and allows the optimization of the single main DC/AC bridge (soft-switching converter), therefore maximizing the overall efficiency. This advantage becomes more significant in case of multiple isolated supplies (such as gate drivers for multilevel structures), as the number of DC/AC units can be reduced dramatically.”

We tried to critically discuss the pros and cons of our solution with respect to a conventional structure (Flyback converter), both to avoid further weighting the paper with an excessive length and for It is quite difficult to quantify the differences among different solutions since the actual design and sizing take a large role in the performance achieved.

Basically, we can argue that the proposed auxiliary power supply solution allows great scalability and flexibility with high robustness to disturbances thanks to the isolation due to the presence of local transformers with quite a reduced number of overall components.

Furthermore, the manageable scalability, of adding LPS in the proposed AC distribution system is discussed both in the Introduction and in the Discussion. Finally, it is recalled in the Conclusions.

The paper writing requires a careful review for correcting grammatical mistakes and typos. There are many figures in this paper, some figures are vague, the manuscript's figures need a brush-up.

We fixed multiple grammatical typos and we improved the quality of the figures.
